# Reparameterizing Mirror Descent as Gradient Descent

**Ehsan Amid**[*] and **Manfred K. Warmuth**
Google Research, Brain Team
Mountain View, CA
{eamid, manfred}@google.com

## Abstract

Most of the recent successful applications of neural networks have been based on training with gradient descent updates. However, for some small networks, other mirror descent updates learn provably more efficiently when the target is sparse. We present a general framework for casting a mirror descent update as a gradient descent update on a different set of parameters. In some cases, the mirror descent reparameterization can be described as training a modified network with standard backpropagation. The reparameterization framework is versatile and covers a wide range of mirror descent updates, even cases where the domain is constrained. Our construction for the reparameterization argument is done for the continuous versions of the updates. Finding general criteria for the discrete versions to closely track their continuous counterparts remains an interesting open problem.

## 1   Introduction

Mirror descent (MD) [Nemirovski and Yudin, 1983, Kivinen and Warmuth, 1997] refers to a family of updates which transform the parameters $\boldsymbol{w} \in \mathcal{C}$ from a convex domain $\mathcal{C} \in \mathbb{R}^d$ via a *link* function (a.k.a. mirror map) $f : \mathcal{C} \to \mathbb{R}^d$ before applying the descent step. The *continuous-time mirror descent* (CMD) update, which can be seen as the limit case of (discrete-time) MD, corresponds to the solution of the following ordinary differential equation (ODE) [Nemirovski and Yudin, 1983, Warmuth and Jagota, 1998, Raginsky and Bouvrie, 2012]:

$$\frac{f(\boldsymbol{w}(t+h)) - f(\boldsymbol{w}(t))}{h} \overset{h \to 0}{=} \dot{f}(\boldsymbol{w}(t)) = -\nabla L(\boldsymbol{w}(t)), \qquad \text{(CMD)} \qquad (1)$$

$$\boldsymbol{w}(t+h) = f^{-1}\Big(f(\boldsymbol{w}(t)) - h \nabla L(\boldsymbol{w}(t))\Big). \qquad \text{(MD)} \qquad (2)$$

Here $L$ denotes a differentiable real-valued loss and $\dot{f} := \frac{\partial f}{\partial t}$ is the time derivative of the link function. The vanilla discretized MD update is obtained by setting the step size to $h$. The main link functions investigated in the past are $f(\boldsymbol{w}) = \boldsymbol{w}$ and $f(\boldsymbol{w}) = \log(\boldsymbol{w})$ leading to the gradient descent (GD) and the unnormalized exponentiated gradient (EGU) family of updates.[2] These two link functions are associated with the squared Euclidean and the relative entropy divergences, respectively. For example, the classical Perceptron and Winnow algorithms are motivated using the identity and log links, respectively, when the loss is the hinge loss. A number of papers discuss the difference between the two updates [Kivinen and Warmuth, 1997, Kivinen et al., 2006, Nie et al., 2016, Ghai et al., 2020] and their rotational invariance properties have been explored in [Warmuth et al., 2014]. In

---

[*]An earlier version of this manuscript (with additional results on the matrix case) appeared as "Interpolating Between Gradient Descent and Exponentiated Gradient Using Reparameterized Gradient Descent" as a preprint.

[2]The normalized version is called EG and the two-sided version EGU$^\pm$. More about this later.

particular, the *Hadamard problem* is a paradigmatic linear problem which shows that EGU can converge dramatically faster than GD when the instances are dense and the target weight vector is sparse [Kivinen et al., 1997, Warmuth and Vishwanathan, 2005]. This property is linked to the strong-convexity of the relative entropy w.r.t. the $L_1$-norm[3] [Shalev-Shwartz et al., 2012], which motivates the discrete EGU update.

**Contributions**    Although other MD updates can be drastically more efficient than GD updates on certain classes of problems, it was assumed that such MD updates are not realizable using GD. In this paper, we show that in fact a large number of MD updates (such as EGU, and those motivated by the Burg and Inverse divergences) can be reparameterized as GD updates. Concretely, our contributions can be summarized as follows.

- We cast continuous MD updates as minimizing a trade off between a *Bregman momentum* and the loss. We also derive the dual, natural gradient, and the constrained versions of the updates.

- We then provide a general framework that allows reparameterizing one CMD update by another. It requires the existence of a certain reparameterization function and a condition on the derivatives of the two link functions as well as the reparameterization function.

- Specifically, we show that on certain problems, the implicit bias of the GD updates can be controlled by considering a family of *tempered* updates (parameterized by a *temperature* $\tau \in \mathbb{R}$) that interpolate between GD (with $\tau = 0$) and EGU (with $\tau = 1$), while covering a wider class of updates.

We conclude the paper with a number of open problems for future research directions.

**Previous work**    There has been an increasing amount of of interest recently in determining the implicit bias of learning algorithms [Gunasekar et al., 2017, 2018, Vaskevicius et al., 2019]. Here, we mainly focus on the MD updates. The special case of reparameterizing continuous EGU as continuous GD was already known [Akin, 1979, Amid and Warmuth, 2020]. In this paper, we develop a more general framework for reparameterizing one CMD update by another. We give a large variety of examples for reparameterizing the CMD updates as continuous GD updates. The main new examples we consider are based on the tempered versions of the relative entropy divergence [Amid et al., 2019]. The main open problem regarding the CMD updates is whether the discretization of the reparameterized updates track the discretization of the original (discretized) MD updates. The strongest methodology for showing this would be to prove the same regret bounds for the discretized reparameterized update as for the original. This has been done in a case-by-case basis for the EG family [Amid and Warmuth, 2020]. For more discussion see the conclusion section, where we also discuss how our reparameterization method allows exploring the effect of the structure of the neural network on the implicit bias.

**Some basic notation**    We use $\odot$, $\oslash$, and superscript $^{\odot}$ for element-wise product, division, and power, respectively. We let $\boldsymbol{w}(t)$ denote the weight or parameter vector as a function of time $t$. Learning proceeds in steps. During step $s$, we start with weight vector $\boldsymbol{w}(sh) = \boldsymbol{w}_s$ and go to $\boldsymbol{w}((s+1)h) = \boldsymbol{w}_{s+1}$ while processing a batch of examples. We also write the Jacobian of vector valued function $q$ as $\boldsymbol{J}_q$ and use $\boldsymbol{H}_F$ to denote the Hessian of a scalar function $F$. Furthermore, we let $\nabla_{\boldsymbol{w}} F(\boldsymbol{w}(t))$ denote the gradient of function $F(\boldsymbol{w})$ evaluated at $\boldsymbol{w}(t)$ and often drop the subscript $\boldsymbol{w}$ for conciseness.

## 2   Continuous-time Mirror Descent

For a strictly convex, continuously-differentiable function $F : \mathcal{C} \to \mathbb{R}$ with convex domain $\mathcal{C} \subseteq \mathbb{R}^d$, the *Bregman divergence* between $\widetilde{\boldsymbol{w}}, \boldsymbol{w} \in \mathcal{C}$ is defined as

$$D_F(\widetilde{\boldsymbol{w}}, \boldsymbol{w}) \coloneqq F(\widetilde{\boldsymbol{w}}) - F(\boldsymbol{w}) - f(\boldsymbol{w})^{\top}(\widetilde{\boldsymbol{w}} - \boldsymbol{w}),$$

where $f \coloneqq \nabla F$ denotes the gradient of $F$, sometimes called the *link function*.[4] Trading off the divergence to the last parameter $\boldsymbol{w}_s$ with the current loss lets us motivate the iterative *mirror descent*

(MD) updates [Nemirovski and Yudin, 1983, Kivinen and Warmuth, 1997]:

$$\boldsymbol{w}_{s+1} = \operatorname*{argmin}_{\boldsymbol{w}} \ {}^{1}\!/\!_{h}\, D_F(\boldsymbol{w}, \boldsymbol{w}_s) + L(\boldsymbol{w}) \,, \tag{3}$$

where $h > 0$ is often called the *learning rate*. Solving for $\boldsymbol{w}_{s+1}$ yields the so-called *prox* or *implicit update* [Rockafellar, 1976, Nemirovski and Yudin, 1983, Kivinen et al., 2006]:

$$f(\boldsymbol{w}_{s+1}) = f(\boldsymbol{w}_s) - h\, \nabla L(\boldsymbol{w}_{s+1}) \,. \tag{4}$$

This update is typically approximated by the following *explicit* update that uses the gradient at the old parameter $\boldsymbol{w}_s$ instead (denoted here as the MD update):

$$f(\boldsymbol{w}_{s+1}) = f(\boldsymbol{w}_s) - h\, \nabla L(\boldsymbol{w}_s) \,. \tag{MD} \tag{5}$$

We now show that the CMD update (1) can be motivated similarly by replacing the Bregman divergence in the minimization problem (3) with a "momentum" version which quantifies the rate of change in the value of Bregman divergence as $\boldsymbol{w}(t)$ varies over time. For the convex function $F$, we define the *Bregman momentum* between $\boldsymbol{w}(t), \boldsymbol{w}_0 \in \mathcal{C}$ as the time differential of the Bregman divergence induced by $F$,

$$\dot{D}_F(\boldsymbol{w}(t), \boldsymbol{w}_0) = \dot{F}(\boldsymbol{w}(t)) - f(\boldsymbol{w}_0)^\top \dot{\boldsymbol{w}}(t) = \big(f(\boldsymbol{w}(t)) - f(\boldsymbol{w}_0)\big)^\top \dot{\boldsymbol{w}}(t) \,.$$

**Theorem 1** (Main result #1). *The CMD update*[5]

$$\dot{f}\big(\boldsymbol{w}(t)\big) = -\nabla L(\boldsymbol{w}(t)) \,, \ \text{with initial condition } \boldsymbol{w}(0) = \boldsymbol{w}_0,$$

*is the solution of the following functional*[6]

$$\min_{\text{curve } \boldsymbol{w}(t)} \ \left\{ \dot{D}_F(\boldsymbol{w}(t), \boldsymbol{w}_0) + L(\boldsymbol{w}(t)) \right\} . \tag{6}$$

*Proof.* Setting the derivatives w.r.t. $\boldsymbol{w}(t)$ to zero, we have

$$\frac{\partial}{\partial \boldsymbol{w}(t)} \Big( \big(f(\boldsymbol{w}(t)) - f(\boldsymbol{w}_0)\big)^\top \dot{\boldsymbol{w}}(t) + L(\boldsymbol{w}(t)) \Big)$$

$$= \boldsymbol{H}_F(\boldsymbol{w}(t))\, \dot{\boldsymbol{w}}(t) + \frac{\partial \dot{\boldsymbol{w}}(t)}{\partial \boldsymbol{w}(t)} \big( f(\boldsymbol{w}(t)) - f(\boldsymbol{w}_0) \big) + \nabla L(\boldsymbol{w}(t))$$

$$= \dot{f}\big(\boldsymbol{w}(t)\big) + \nabla L(\boldsymbol{w}(t)) = \boldsymbol{0} \,,$$

where we use the fact that $\boldsymbol{w}(t)$ and $\dot{\boldsymbol{w}}(t)$ are independent variables[7] [Burke, 1985], thus $\frac{\partial \dot{\boldsymbol{w}}(t)}{\partial \boldsymbol{w}(t)} = \boldsymbol{0}$. □

Note that the implicit update (4) and the explicit update (5) can both be realized as the backward and the forward Euler approximations of (1), respectively, with step size $h$. Alternatively, (3) can be obtained from (6) via a simple discretization of the momentum term (see Appendix C).

We can provide an alternative definition of Bregman momentum in terms of the dual of $F$ function. If $F^*(\boldsymbol{w}^*) = \sup_{\widetilde{\boldsymbol{w}} \in \mathcal{C}} \big(\widetilde{\boldsymbol{w}}^\top \boldsymbol{w}^* - F(\widetilde{\boldsymbol{w}})\big)$ denotes the Fenchel dual of $F$ and $\boldsymbol{w} = \arg\sup_{\widetilde{\boldsymbol{w}} \in \mathcal{C}} (\widetilde{\boldsymbol{w}}^\top \boldsymbol{w}^* - F(\widetilde{\boldsymbol{w}}))$, then the following relation holds between the pair of dual variables $(\boldsymbol{w}, \boldsymbol{w}^*)$:

$$\boldsymbol{w} = f^*(\boldsymbol{w}^*) \,, \quad \boldsymbol{w}^* = f(\boldsymbol{w}) \,, \quad \text{and} \quad f^* = f^{-1} \,. \tag{7}$$

Taking the derivative of $\boldsymbol{w}(t)$ and $\boldsymbol{w}^*(t)$ w.r.t. $t$ yields:

$$\dot{\boldsymbol{w}}(t) = \dot{f}^*\big(\boldsymbol{w}^*(t)\big) = \boldsymbol{H}_{F^*}\big(\boldsymbol{w}^*(t)\big) \dot{\boldsymbol{w}}^*(t) \,, \tag{8} \qquad \dot{\boldsymbol{w}}^*(t) = \dot{f}\big(\boldsymbol{w}(t)\big) = \boldsymbol{H}_F\big(\boldsymbol{w}(t)\big) \dot{\boldsymbol{w}}(t) \,. \tag{9}$$

This pairing allows rewriting the Bregman momentum in its dual form:

$$\dot{D}_F(\boldsymbol{w}(t), \boldsymbol{w}_0) = \dot{D}_{F^*}(\boldsymbol{w}_0^*, \boldsymbol{w}^*(t)) = (\boldsymbol{w}^*(t) - \boldsymbol{w}_0^*)^\top \boldsymbol{H}_{F^*}(\boldsymbol{w}^*(t)) \ \dot{\boldsymbol{w}}^*(t) \,. \tag{10}$$

An expanded derivation is given in Appendix A. Using (9), we can rewrite the CMD update (1) as

$$\dot{\boldsymbol{w}}(t) = -\boldsymbol{H}_F^{-1}(\boldsymbol{w}(t))\,\nabla L(\boldsymbol{w}(t))\,, \qquad\text{(NGD)} \qquad (11)$$

i.e. a natural gradient descent (NGD) update [Amari, 1998] w.r.t. the Riemannian metric $\boldsymbol{H}_F$. Using $\nabla L(\boldsymbol{w}) = \boldsymbol{H}_{F^*}(\boldsymbol{w}^*)\nabla_{\boldsymbol{w}^*} L\circ f^*(\boldsymbol{w}^*)$ and $\boldsymbol{H}_F(\boldsymbol{w}) = \boldsymbol{H}_{F^*}^{-1}(\boldsymbol{w}^*)$, the CMD update (1) can be written equivalently in the dual domain $\boldsymbol{w}^*$ as an NGD update w.r.t. the Riemannian metric $\boldsymbol{H}_{F^*}$, or by applying (8) as a CMD with the link $f^*$:

$$\dot{\boldsymbol{w}}^*(t) = -\boldsymbol{H}_{F^*}^{-1}(\boldsymbol{w}^*(t))\,\nabla_{\boldsymbol{w}^*} L\circ f^*(\boldsymbol{w}^*(t))\,, \quad (12) \quad \dot{f}^*(\boldsymbol{w}^*(t)) = -\nabla_{\boldsymbol{w}^*} L\circ f^*(\boldsymbol{w}^*(t))\,. \quad (13)$$

The equivalence of the primal-dual updates was already shown in [Warmuth and Jagota, 1998] for the continuous case and in [Raskutti and Mukherjee, 2015] for the discrete case (where it only holds in one direction). We will show that the equivalence relation is a special case of the reparameterization theorem, introduced in the next section. In the following, we discuss the projected CMD updates for the constrained setting.

**Proposition 1.** *The CMD update with the additional constraint $\psi\big(\boldsymbol{w}(t)\big) = \boldsymbol{0}$ for some function $\psi : \mathbb{R}^d \to \mathbb{R}^m$ s.t. $\{\boldsymbol{w} \in \mathcal{C}\,|\,\psi\big(\boldsymbol{w}(t)\big) = \boldsymbol{0}\}$ is non-empty, amounts to the projected gradient update*

$$\dot{f}\big(\boldsymbol{w}(t)\big) = -\boldsymbol{P}_\psi(\boldsymbol{w}(t))\nabla L(\boldsymbol{w}(t)) \;\&\; \dot{f}^*(\boldsymbol{w}^*(t)) = -\boldsymbol{P}_\psi(\boldsymbol{w}(t))^\top\,\nabla L\circ f^*\,(\boldsymbol{w}^*(t))\,, \qquad (14)$$

*where $\boldsymbol{P}_\psi \coloneqq \boldsymbol{I}_d - \boldsymbol{J}_\psi^\top\big(\boldsymbol{J}_\psi \boldsymbol{H}_F^{-1}\boldsymbol{J}_\psi^\top\big)^{-1}\boldsymbol{J}_\psi \boldsymbol{H}_F^{-1}$ is the projection matrix onto the tangent space of $F$ at $\boldsymbol{w}(t)$ and $\boldsymbol{J}_\psi(\boldsymbol{w}(t))$. Equivalently, the update can be written as a projected natural gradient descent update*

$$\dot{\boldsymbol{w}}(t)=-\boldsymbol{P}_\psi^\top(\boldsymbol{w}(t))\boldsymbol{H}_F^{-1}(\boldsymbol{w}(t))\nabla L(\boldsymbol{w}(t)) \;\&\; \dot{\boldsymbol{w}}^*(t)=-\boldsymbol{P}_\psi \boldsymbol{H}_{F^*}^{-1}(\boldsymbol{w}^*(t))\nabla L\circ f^*(\boldsymbol{w}^*(t)). \quad (15)$$

**Example 1** ((Normalized) EG). *The unnormalized EG update is motivated using the link function $f(\boldsymbol{w}) = \log \boldsymbol{w}$. Adding the linear constraint $\psi(\boldsymbol{w}) = \boldsymbol{w}^\top \boldsymbol{1} - 1$ to the unnormalized EG update results in the (normalized) EG update [Kivinen and Warmuth, 1997]. Since $\boldsymbol{J}_\psi(\boldsymbol{w}) = \boldsymbol{1}^\top$ and $\boldsymbol{H}_F(\boldsymbol{w})^{-1} = \mathrm{diag}(\boldsymbol{w})$, $\boldsymbol{P}_\psi = \boldsymbol{I} - \frac{\boldsymbol{1}\boldsymbol{1}^\top \mathrm{diag}(\boldsymbol{w})}{\boldsymbol{1}^\top \mathrm{diag}(\boldsymbol{w})\boldsymbol{1}} = \boldsymbol{I} - \boldsymbol{1}\boldsymbol{w}^\top$ and the projected CMD update (15) (the continuous EG update) and its NGD form become*

$$\dot{\log}(\boldsymbol{w}) = -(\boldsymbol{I} - \boldsymbol{1}\boldsymbol{w}^\top)\,\nabla L(\boldsymbol{w}) = -(\nabla L(\boldsymbol{w}) - \boldsymbol{1}\,\boldsymbol{w}^\top\nabla L(\boldsymbol{w}))\,,$$

$$\dot{\boldsymbol{w}} = -(\mathrm{diag}(\boldsymbol{w})\nabla L(\boldsymbol{w}) - \boldsymbol{w}\,\boldsymbol{w}^\top\nabla L(\boldsymbol{w}))\,.$$

## 3  Reparameterization

We now establish the second main result of the paper.

**Theorem 2** (Main result #2). *Let $F$ and $G$ be strictly convex, continuously-differentiable functions with domains in $\mathbb{R}^d$ and $\mathbb{R}^k$, respectively, s.t. $k \geq d$. Let $q : \mathbb{R}^k \to \mathbb{R}^d$ be a reparameterization function expressing parameters $\boldsymbol{w}$ of $F$ uniquely as $q(\boldsymbol{u})$ where $\boldsymbol{u}$ lies in the domain of $G$. Then the CMD update on parameter $\boldsymbol{w}$ for the convex function $F$ (with link $f(\boldsymbol{w}) = \nabla F(\boldsymbol{w})$) and loss $L(\boldsymbol{w})$,*

$$\dot{f}(\boldsymbol{w}(t)) = -\nabla L(\boldsymbol{w}(t))\,,$$

*coincides with the CMD update on parameters $\boldsymbol{u}$ for the convex function $G$ (with link $g(\boldsymbol{u}) \coloneqq \nabla G(\boldsymbol{u})$) and the composite loss $L\circ q$,*

$$\dot{g}(\boldsymbol{u}(t)) = -\nabla_{\boldsymbol{u}} L\circ q\big(\boldsymbol{u}(t)\big)\,,$$

*provided that $\boldsymbol{w}(0) = q(\boldsymbol{u}(0))$ and $\mathrm{range}(q) \subseteq \mathrm{dom}(F)$ hold, and we have*

$$\boldsymbol{H}_F^{-1}(\boldsymbol{w}) = \boldsymbol{J}_q(\boldsymbol{u})\,\boldsymbol{H}_G^{-1}(\boldsymbol{u})\,\boldsymbol{J}_q(\boldsymbol{u})^\top\,, \text{ for all } \boldsymbol{w} = q(\boldsymbol{u})\,.$$

*Proof.* Note that (dropping $t$ for simplicity) we have $\dot{\boldsymbol{w}} = \frac{\partial \boldsymbol{w}}{\partial \boldsymbol{u}}\,\dot{\boldsymbol{u}} = \boldsymbol{J}_q(\boldsymbol{u})\,\dot{\boldsymbol{u}}$ and $\nabla_{\boldsymbol{u}} L\circ q(\boldsymbol{u}) = \boldsymbol{J}_q(\boldsymbol{u})^\top \nabla L(\boldsymbol{w})$. The CMD update on $\boldsymbol{u}$ with the link function $g(\boldsymbol{u})$ can be written in the NGD form as $\dot{\boldsymbol{u}} = -\boldsymbol{H}_G^{-1}(\boldsymbol{u})\nabla_{\boldsymbol{u}} L\circ q(\boldsymbol{u})$. Thus,

$$\dot{\boldsymbol{u}} = -\boldsymbol{H}_G^{-1}(\boldsymbol{u})\,\boldsymbol{J}_q(\boldsymbol{u})^\top\,\nabla_{\boldsymbol{w}} L(\boldsymbol{w})\,.$$

Multiplying by $\boldsymbol{J}_q(\boldsymbol{u})$ from the left yields

$$\dot{\boldsymbol{w}} = -\boldsymbol{J}_q(\boldsymbol{u})\boldsymbol{H}_G^{-1}(\boldsymbol{u})\boldsymbol{J}_q(\boldsymbol{u})^\top\nabla_{\boldsymbol{w}} L(\boldsymbol{w})\,.$$

Comparing the result to (11) concludes the proof. □

In the following examples, we will mainly consider reparameterizing a CMD update with the link function $f(\boldsymbol{w})$ as a GD update on $\boldsymbol{u}$, for which we have $\boldsymbol{H}_G = \boldsymbol{I}_k$.

**Example 2** (EGU as GD). *The continuous-time EGU can be reparameterized as continuous GD with the reparameterization function $\boldsymbol{w} = q(\boldsymbol{u}) = {}^1\!/\!_4\,\boldsymbol{u} \odot \boldsymbol{u} = {}^1\!/\!_4\,\boldsymbol{u}^{\odot 2}$, i.e.*

$$\dot{\overline{\log(\boldsymbol{w})}} = -\nabla L(\boldsymbol{w}) \;\; equals \;\; \dot{\boldsymbol{u}} = -\underbrace{\nabla L \circ q\,(\boldsymbol{u})}_{\nabla_{\boldsymbol{u}} L\,({}^1\!/\!_4\,\boldsymbol{u}^{\odot 2})} = -{}^1\!/\!_2\,\boldsymbol{u} \odot \nabla L(\boldsymbol{w})\,.$$

*This is proven by verifying the condition of Theorem 2:*

$$\boldsymbol{J}_q(\boldsymbol{u})\boldsymbol{J}_q(\boldsymbol{u})^\top = {}^1\!/\!_2 \operatorname{diag}(\boldsymbol{u})\,({}^1\!/\!_2 \operatorname{diag}(\boldsymbol{u}))^\top = \operatorname{diag}({}^1\!/\!_4\,\boldsymbol{u}^{\odot 2}) = \operatorname{diag}(\boldsymbol{w}) = \boldsymbol{H}_F^{-1}(\boldsymbol{w})\,.$$

**Example 3** (Reduced EG in 2-dimension). *Consider the 2-dimensional normalized weights $\boldsymbol{w} = [\omega, 1-\omega]^\top$ where $0 \le \omega \le 1$. The normalized reduced EG update [Warmuth and Jagota, 1998] is motivated by the link function $f(w) = \log \frac{w}{1-w}$, thus $H_F(w) = \frac{1}{w} + \frac{1}{1-w} = \frac{1}{w(1-w)}$. This update can be reparameterized as a GD update on $u \in \mathbb{R}$ via $\omega = q(u) = {}^1\!/\!_2(1 + \sin(u))$ i.e.*

$$\dot{\overline{\log(\frac{w}{1-w})}} = -\nabla_w L(w) \;\; equals \;\; \dot{u} = -\underbrace{\nabla_u L \circ q\,(u)}_{\nabla_u L\,\left({}^1\!/\!_2(1+\sin(u))\right)} = -\frac{\cos(u)}{2}\,\nabla L(w)\,.$$

*This is verified by checking the condition of Theorem 2: $J_q(u) = {}^1\!/\!_2 \cos(u)$ and*

$$J_q(u) J_q(u)^\top = \frac{1}{4}\,\cos^2(u) = \frac{1}{2}\left(1 + \sin(u)\right) \frac{1}{2}\left(1 - \sin(u)\right) = w(1-w) = H_F^{-1}(w)\,.$$

**Open problem** The generalization of the reduced EG link function to $d > 2$ dimensions becomes $f(\boldsymbol{w}) = \log \frac{\boldsymbol{w}}{1 - \sum_{i=1}^{d-1} w_i}$ which utilizes the first $(d-1)$-dimensions $\boldsymbol{w}$ s.t. $[\boldsymbol{w}^\top, w_d]^\top \in \Delta^{d-1}$. Reparameterizing the CMD update using this link as CGD is open. The update can be reformulated as

$$\dot{\boldsymbol{w}} = -\Big(\operatorname{diag}\big(\boldsymbol{1} \oslash \boldsymbol{w}\big) + \frac{1}{1 - \sum_{i=1}^{d-1} w_i}\,\boldsymbol{1}\boldsymbol{1}^\top\Big)^{-1} \nabla L(\boldsymbol{w}) = -\big(\operatorname{diag}(\boldsymbol{w}) - \boldsymbol{w}\boldsymbol{w}^\top\big) \nabla L(\boldsymbol{w})\,.$$

We will give a $d$-dimensional version of EG using a projection onto a constraint in Example 6.

**Example 4** (Burg updates as GD). *The update associated with the negative Burg entropy $F(\boldsymbol{w}) = -\sum_{i=1}^d \log w_i$ and link $f(\boldsymbol{w}) = -\boldsymbol{1} \oslash \boldsymbol{w}$ is reparameterized as GD with $\boldsymbol{w} = q(\boldsymbol{u}) := \exp(\boldsymbol{u})$, i.e.*

$$\dot{\overline{(-\boldsymbol{1} \oslash \boldsymbol{w})}} = -\nabla L(\boldsymbol{w}) \;\; equals \;\; \dot{\boldsymbol{u}} = -\underbrace{\nabla L \circ q\,(\boldsymbol{u})}_{\nabla_{\boldsymbol{u}} L\,(\exp(\boldsymbol{u}))} = -\exp(\boldsymbol{u}) \odot \nabla L(\boldsymbol{w})\,,$$

*This is verified by the condition of Theorem 2: $\boldsymbol{H}_F(\boldsymbol{w}) = \operatorname{diag}(\boldsymbol{1} \oslash \boldsymbol{w})^2$, $\boldsymbol{J}_q(\boldsymbol{u}) = \operatorname{diag}(\exp(\boldsymbol{u}))$, and*

$$\boldsymbol{J}_q(\boldsymbol{u})\boldsymbol{J}_q(\boldsymbol{u})^\top = \operatorname{diag}(\exp(\boldsymbol{u}))^2 = \operatorname{diag}(\boldsymbol{w})^2 = \boldsymbol{H}_F^{-1}(\boldsymbol{w})\,.$$

**Example 5** (EGU as Burg). *The reparameterization step can be chained, and applied in reverse, when the reparameterization function $q$ is invertible. For instance, we can first apply the inverse reparameterization of the Burg update as GD from Example 4, i.e. $\boldsymbol{u} = q^{-1}(\boldsymbol{w}) = \log \boldsymbol{w}$. Subsequently, applying the reparameterization of EGU as GD from Example 2, i.e. $\boldsymbol{v} = \tilde{q}(\boldsymbol{u}) = {}^1\!/\!_4\,\boldsymbol{u}^{\odot 2}$, results in the reparameterization of EGU update on $\boldsymbol{v}$ as Burg update on $\boldsymbol{w}$, that is,*

$$\dot{\overline{\log(\boldsymbol{v})}} = -\nabla L(\boldsymbol{v}) \;\; equals \;\; \dot{\overline{(-\boldsymbol{1} \oslash \boldsymbol{w})}} = -\underbrace{\nabla_{\boldsymbol{w}} L \circ \tilde{q} \circ q^{-1}(\boldsymbol{w})}_{\nabla_{\boldsymbol{w}} L({}^1\!/\!_4(\log \boldsymbol{w})^{\odot 2})} = -(\log(\boldsymbol{w}) \oslash (2\boldsymbol{w})) \odot \nabla L(\boldsymbol{v})\,.$$

For completeness, we also provide the constrained reparameterized updates (proof in Appendix B).

**Theorem 3.** *The constrained CMD update (14) coincides with the reparameterized projected gradient update on the composite loss,*

$$\dot{\overline{g(\boldsymbol{u}(t))}} = -\boldsymbol{P}_{\psi \circ q}(\boldsymbol{u}(t)) \nabla_{\boldsymbol{u}} L \circ q(\boldsymbol{u}(t))\,,$$

*where $\boldsymbol{P}_{\psi \circ q} := \boldsymbol{I}_k - \boldsymbol{J}_{\psi \circ q}^\top\big(\boldsymbol{J}_{\psi \circ q} \boldsymbol{H}_G^{-1} \boldsymbol{J}_{\psi \circ q}^\top\big)^{-1} \boldsymbol{J}_{\psi \circ q} \boldsymbol{H}_G^{-1}$ is the projection matrix onto the tangent space at $\boldsymbol{u}(t)$ and $\boldsymbol{J}_{\psi \circ q}(\boldsymbol{u}) := \boldsymbol{J}_q^\top(\boldsymbol{u}) \boldsymbol{J}_\psi(\boldsymbol{w})$.*

**Example 6** (EG as GD). *We now extend the reparameterization of the EGU update as GD in Example 2 to the normalized case in terms of a projected GD update. Combining $q(\boldsymbol{u}) = 1/4\,\boldsymbol{u}^{\odot 2}$ with $\psi(\boldsymbol{w}) = \mathbf{1}^{\top}\boldsymbol{w} - 1$, we have $\boldsymbol{J}_{\psi \circ q}(\boldsymbol{u}) = 1/2 \operatorname{diag}(\boldsymbol{u})\,\mathbf{1}^{\top} = 1/2\,\boldsymbol{u}^{\top}$ and $\boldsymbol{P}_{\psi \circ q}(\boldsymbol{u}) = \boldsymbol{I} - \frac{1/4\,\boldsymbol{u}\boldsymbol{u}^{\top}}{1/4\|\boldsymbol{u}\|^2} = \boldsymbol{I} - 1/4\,\boldsymbol{u}\boldsymbol{u}^{\top}$. Thus,*

$$\dot{\boldsymbol{u}} = -\big(\boldsymbol{I} - 1/4\,\boldsymbol{u}\boldsymbol{u}^{\top}\big)\nabla_{\boldsymbol{u}}L(1/4\,\boldsymbol{u}^{\odot 2}) \text{ with } \boldsymbol{w}(t) = 1/4\,\boldsymbol{u}(t)^{\odot 2},$$

*equals the normalized EG update in Example 2. Note that similar ideas were explored in an evolutionary game theory context in [Sandholm, 2010].*

## 4 Tempered Updates

In this section, we consider a richer class of examples derived using the tempered relative entropy divergence [Amid et al., 2019], parameterized by a *temperature* $\tau \in \mathbb{R}$. As we will see, the tempered updates allow interpolating between many well-known cases. We start with the tempered logarithm link function [Naudts, 2002]:

$$f_{\tau}(\boldsymbol{w}) = \log_{\tau}(\boldsymbol{w}) = \frac{1}{1-\tau}(\boldsymbol{w}^{1-\tau} - 1), \qquad (16)$$

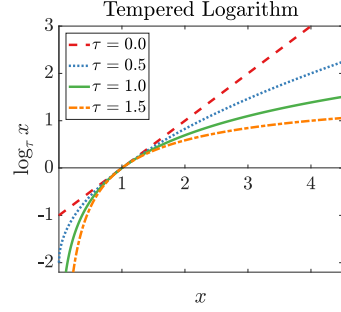

Tempered Logarithm

Figure 1: $\log_{\tau}(x)$, for different $\tau \geq 0$.

for $\boldsymbol{w} \in \mathbb{R}^d_{\geq 0}$ and $\tau \in \mathbb{R}$. The $\log_{\tau}$ function is shown in Figure 1 for different values of $\tau \geq 0$. Note that $\tau = 1$ recovers the standard log function as a limit point. The $\log_{\tau}(\boldsymbol{w})$ link function is the gradient of the convex function

$$F_{\tau}(\boldsymbol{w}) = \sum_i \Big(w_i \log_{\tau} w_i + \frac{1}{2-\tau}(1 - w_i^{2-\tau})\Big) = \sum_i \Big(\frac{1}{(1-\tau)(2-\tau)}w_i^{2-\tau} - \frac{1}{1-\tau}w_i + \frac{1}{2-\tau}\Big).$$

The convex function $F_{\tau}$ induces the following tempered Bregman divergence[8]:

$$D_{F_{\tau}}(\widetilde{\boldsymbol{w}}, \boldsymbol{w}) = \sum_i \Big(\widetilde{w}_i \log_{\tau} \widetilde{w}_i - \widetilde{w}_i \log_{\tau} w_i - \frac{\widetilde{w}_i^{2-\tau} - w_i^{2-\tau}}{2-\tau}\Big)$$

$$= \frac{1}{1-\tau}\sum_i \Big(\frac{\widetilde{w}_i^{2-\tau} - w_i^{2-\tau}}{2-\tau} - (\widetilde{w}_i - w_i)\,w_i^{1-\tau}\Big). \qquad (17)$$

For $\tau = 0$, we obtain the squared Euclidean divergence $D_{F_0}(\widetilde{\boldsymbol{w}}, \boldsymbol{w}) = \frac{1}{2}\|\widetilde{\boldsymbol{w}} - \boldsymbol{w}\|_2^2$ and for $\tau = 1$, the relative entropy $D_{F_1}(\widetilde{\boldsymbol{w}}, \boldsymbol{w}) = \sum_i(\widetilde{w}_i \log(\widetilde{w}_i/w_i) - \widetilde{w}_i + w_i)$ (See [Amid et al., 2019] for an extensive list of examples).

In the following, we derive the CMD updates using the time derivative of (17) as the tempered Bregman momentum. Notice that the link function $\log_{\tau}(x)$ is only defined for $x \geq 0$ when $\tau > 0$. In order to have a weight $\boldsymbol{w} \in \mathbb{R}^d$, we use the $\pm$-trick [Kivinen and Warmuth, 1997] by maintaining two non-negative weights $\boldsymbol{w}_+$ and $\boldsymbol{w}_-$ and setting $\boldsymbol{w} = \boldsymbol{w}_+ - \boldsymbol{w}_-$. We call this the *tempered EGU$_\tau^\pm$* updates, which contain the standard EGU$^\pm$ updates as a special case of $\tau = 1$. As our final main result, we show that that continuous tempered EGU$_\tau^\pm$ updates interpolate between continuous-time GD and continuous EGU (for $\tau \in [0, 1]$). Furthermore, these updates can be simulated by continuous GD on a new set of parameters $\boldsymbol{u}$ using a simple reparameterization. We show that reparameterizing the tempered updates as GD updates on the composite loss $L \circ q$ changes the implicit bias of GD, making the updates converge to the solution with the smallest $L_{2-\tau}$-norm for arbitrary $\tau \in [0, 1]$.

### 4.1 Tempered EGU and Reparameterization

We first introduce the generalization of the EGU update using the tempered Bregman divergence (17). Let $\boldsymbol{w}(t) \in \mathbb{R}^d_{\geq 0}$. The tempered EGU update is motivated by

$$\operatorname*{argmin}_{\text{curve } \boldsymbol{w}(t) \in \mathbb{R}^d_{\geq 0}} \Big\{ \dot{D}_{F_{\tau}}\big(\boldsymbol{w}(t), \boldsymbol{w}_0\big) + L(\boldsymbol{w}(t)) \Big\}.$$

This results in the CMD update

$$\dot{\overline{\log_{\tau}}}\boldsymbol{w}(t) = -\nabla L(\boldsymbol{w}(t)). \qquad (18)$$

An equivalent integral version of this update is

$$\boldsymbol{w}(t) = \exp_\tau\big(\log_\tau \boldsymbol{w}_0 - \int_0^t \nabla_{\boldsymbol{w}} L(\boldsymbol{w}(z))\,\mathrm{d}z\big), \qquad (19)$$

where $\exp_\tau(x) := [1 + (1-\tau)x]_+^{\frac{1}{1-\tau}}$ is the inverse of tempered logarithm (16). Note that $\tau = 1$ is a limit case which recovers the standard $\exp$ function and the update (18) becomes the standard EGU update. Additionally, the GD update (on the non-negative orthant) is recovered at $\tau = 0$. As a result, the tempered $\text{EGU}_\tau$ update (18) interpolates between GD and EGU for $\tau \in [0, 1]$ and generalizes beyond for values of $\tau > 1$ and $\tau < 0$.[9] We now show the reparameterization of the tempered $\text{EGU}_\tau$ update (18) as GD. This corresponds to continuous-time gradient descent on the network of Figure 2.

**Proposition 2** (Main result #3). *The tempered continuous $\text{EGU}_\tau$ update can be reparameterized continuous-time GD with the reparameterization function*

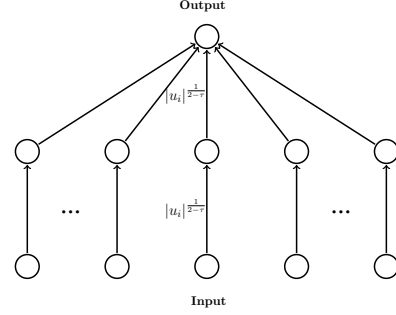

Figure 2: A reparameterized linear neuron where $w_i = |u_i|^{\frac{2}{2-\tau}}$ as a two-layer sparse network: value of $\tau = 0$ reduces to GD while $\tau = 1$ simulates the EGU update.

$$\boldsymbol{w} = q_\tau(\boldsymbol{u}) = \big(\frac{2-\tau}{2}\big)^{\frac{2}{2-\tau}} |\boldsymbol{u}|^{\odot\frac{2}{2-\tau}}\,, \text{ for } \boldsymbol{u} \in \mathbb{R}^d \text{ and } \tau \neq 2\,. \qquad (20)$$

*That is*

$$\dot{\overline{\log_\tau}}(\boldsymbol{w}) = -\nabla L(\boldsymbol{w}) \text{ equals } \dot{\boldsymbol{u}} = -\underbrace{\nabla L \circ q_\tau(\boldsymbol{u})}_{\nabla_{\boldsymbol{u}} L\left(\left(\frac{2-\tau}{2}\right)^{\frac{2}{2-\tau}}|\boldsymbol{u}|^{\odot\frac{2}{2-\tau}}\right)} = -\operatorname{sign}(\boldsymbol{u}) \odot \big(\frac{2-\tau}{2}\big)^{\frac{\tau}{2-\tau}} |\boldsymbol{u}|^{\odot\frac{\tau}{2-\tau}} \odot \nabla L(\boldsymbol{w}).$$

*Proof.* This is verified by checking the condition of Theorem 2. The lhs is

$$(\boldsymbol{H}_{F_\tau(\boldsymbol{w})}(\boldsymbol{w}))^{-1} = (\boldsymbol{J}_{\log_\tau}(\boldsymbol{w}))^{-1} = (\operatorname{diag}(\boldsymbol{w})^{-\tau})^{-1} = \operatorname{diag}(\boldsymbol{w})^\tau.$$

Note that the Jacobian of $q_\tau$ is

$$\boldsymbol{J}_{q_\tau}(\boldsymbol{u}) = \big(\frac{2-\tau}{2}\big)^{\frac{\tau}{2-\tau}} \operatorname{diag}\big(\operatorname{sign}(\boldsymbol{u}) \odot |\boldsymbol{u}|^{\odot\frac{\tau}{2-\tau}}\big) = \operatorname{diag}(\operatorname{sign}(\boldsymbol{u}) \odot q_\tau(\boldsymbol{u})^{\odot\frac{\tau}{2}}).$$

Thus the rhs $\boldsymbol{J}_{q_\tau}(\boldsymbol{u})\boldsymbol{J}_{q_\tau}^\top(\boldsymbol{u})$ of the condition equals $\operatorname{diag}\big(\boldsymbol{w}^{\odot\tau}\big)$ as well. $\qquad\square$

## 4.2 Minimum-norm Solutions

We apply the (reparameterized) tempered $\text{EGU}_\tau^\pm$ update on the underdetermined linear regression problem. For this, we first consider the $\pm$-trick on (18), in which we set $\boldsymbol{w}(t) = \boldsymbol{w}_+(t) - \boldsymbol{w}_-(t)$ where

$$\dot{\overline{\log_\tau}}\boldsymbol{w}_+(t) = -\nabla_{\boldsymbol{w}} L(\boldsymbol{w}(t))\,, \quad \dot{\overline{\log_\tau}}\boldsymbol{w}_-(t) = +\nabla_{\boldsymbol{w}} L(\boldsymbol{w}(t))\,. \qquad (21)$$

Note that using the $\pm$-trick, we have $\boldsymbol{w}(t) \in \mathbb{R}^n$. We call the updates (21) the *tempered $\text{EGU}_\tau^\pm$*. The reparameterization of the tempered $\text{EGU}_\tau^\pm$ updates as GD can be written by applying Proposition 2,

$$\dot{\boldsymbol{u}}_+(t) = -\nabla_{\boldsymbol{u}_+} L\big(q_\tau(\boldsymbol{u}_+(t)) - q_\tau(\boldsymbol{u}_-(t))\big), \dot{\boldsymbol{u}}_-(t) = -\nabla_{\boldsymbol{u}_-} L\big(q_\tau(\boldsymbol{u}_+(t)) - q_\tau(\boldsymbol{u}_-(t))\big), \quad (22)$$

and setting $\boldsymbol{w}(t) = q_\tau(\boldsymbol{u}_+(t)) - q_\tau(\boldsymbol{u}_-(t))$.

The strong convexity of the $F_\tau$ function w.r.t. the $\text{L}_{2-\tau}$-norm (see [Amid et al., 2019]) suggests that the updates motivated by the tempered Bregman divergence (17) yield the minimum $\text{L}_{2-\tau}$-norm solution in certain settings. We verify this by considering the following underdetermined linear regression problem. Let $\{\boldsymbol{x}_n, y_n\}_{n=1}^N$ where $\boldsymbol{x}_n \in \mathbb{R}^d$, $y_n \in \mathbb{R}$ denote the set of input-output pairs and let $\boldsymbol{X} \in \mathbb{R}^{N \times d}$ with $N < d$ be the design matrix for which the $n$-th row is equal to $\boldsymbol{x}_n^\top$. Also, let $\boldsymbol{y} \in \mathbb{R}^N$ denote the vector of targets. Consider the tempered $\text{EGU}_\tau^\pm$ updates (21) on the weights $\boldsymbol{w}(t) = \boldsymbol{w}_+(t) - \boldsymbol{w}_-(t)$ where $\boldsymbol{w}_+(t), \boldsymbol{w}_-(t) \geq \boldsymbol{0}$ and $\boldsymbol{w}_+(0) = \boldsymbol{w}_-(0) = \boldsymbol{w}_0$. Following (19), we have

$$\boldsymbol{w}_+(t) = \exp_\tau\big(\log_\tau \boldsymbol{w}_0 - \int_0^t \boldsymbol{X}^\top \boldsymbol{\delta}(z)\,\mathrm{d}z\big)\,, \quad \boldsymbol{w}_-(t) = \exp_\tau\big(\log_\tau \boldsymbol{w}_0 + \int_0^t \boldsymbol{X}^\top \boldsymbol{\delta}(z)\,\mathrm{d}z\big)\,,$$

where $\boldsymbol{\delta}(t) = \boldsymbol{X}\big(\boldsymbol{w}_+(t) - \boldsymbol{w}_-(t)\big)\,.$

**Theorem 4.** *Consider the underdetermined linear regression problem where $N < d$. Let $\mathcal{E} = \{\boldsymbol{w} \in \mathbb{R}^d \,|\, \boldsymbol{X}\boldsymbol{w} = \boldsymbol{y}\}$ be the set of solutions with zero error. Given $\boldsymbol{w}(\infty) \in \mathcal{E}$, then the tempered $EGU_\tau^\pm$ updates (21) with temperature $0 \le \tau \le 1$ and initial solution $\boldsymbol{w}_0 = \alpha\boldsymbol{1} \ge \boldsymbol{0}$ converge to the minimum $L_{2-\tau}$-norm solution in $\mathcal{E}$ in the limit $\alpha \to 0$.*

*Proof.* We show that the solution of the tempered $EGU_\tau^\pm$ satisfies the dual feasibility and complementary slackness KKT conditions for the following optimization problem (omitting $t$ for simplicity):

$$\min_{\boldsymbol{w}_+, \boldsymbol{w}_-} \|\boldsymbol{w}_+ - \boldsymbol{w}_-\|_{2-\tau}^{2-\tau}, \text{ for } 0 \le \tau \le 1, \quad \text{s.t.} \quad \boldsymbol{X}(\boldsymbol{w}_+ - \boldsymbol{w}_-) = \boldsymbol{y} \text{ and } \boldsymbol{w}_+, \boldsymbol{w}_- \ge \boldsymbol{0}.$$

Imposing the constraints using a set of Lagrange multipliers $\boldsymbol{\nu}_+, \boldsymbol{\nu}_- \ge \boldsymbol{0}$ and $\boldsymbol{\lambda} \in \mathbb{R}$, we have

$$\min_{\boldsymbol{w}} \sup_{\boldsymbol{\nu}_+, \boldsymbol{\nu}_- \ge \boldsymbol{0}, \boldsymbol{\lambda}} \left\{ \|\boldsymbol{w}_+ - \boldsymbol{w}_-\|_{2-\tau}^{2-\tau} + \boldsymbol{\lambda}^\top \left( \boldsymbol{X}(\boldsymbol{w}_+ - \boldsymbol{w}_-) - \boldsymbol{y} \right) - \boldsymbol{w}_+^\top \boldsymbol{\nu}_+ - \boldsymbol{w}_-^\top \boldsymbol{\nu}_- \right\}.$$

The set of KKT conditions are

$$\begin{cases} \boldsymbol{w}_+, \boldsymbol{w}_- \ge \boldsymbol{0}, \; \boldsymbol{X}\boldsymbol{w} = \boldsymbol{y}, \\ + \operatorname{sign}(\boldsymbol{w}) \odot |\boldsymbol{w}|^{\odot(1-\tau)} - \boldsymbol{X}^\top \boldsymbol{\lambda} \ge \boldsymbol{0}, \; -\operatorname{sign}(\boldsymbol{w}) \odot |\boldsymbol{w}|^{\odot(1-\tau)} + \boldsymbol{X}^\top \boldsymbol{\lambda} \ge \boldsymbol{0}, \\ \left( \operatorname{sign}(\boldsymbol{w}) \odot |\boldsymbol{w}|^{\odot(1-\tau)} - \boldsymbol{X}^\top \boldsymbol{\lambda} \right) \odot \boldsymbol{w}_+ = \boldsymbol{0}, \; \left( \operatorname{sign}(\boldsymbol{w}) \odot |\boldsymbol{w}|^{\odot(1-\tau)} - \boldsymbol{X}^\top \boldsymbol{\lambda} \right) \odot \boldsymbol{w}_- = \boldsymbol{0}, \end{cases}$$

where $\boldsymbol{w} = \boldsymbol{w}_+ - \boldsymbol{w}_-$. The first condition is imposed by the form of the updates and the second condition is satisfied by the assumption at $t \to \infty$. Using $\boldsymbol{w}_0 = \alpha\boldsymbol{1}$ with $\alpha \to 0$, we have

$$\boldsymbol{w}_+(t) = \exp_\tau \left( -\frac{1}{1-\tau} - \int_0^t \boldsymbol{X}^\top \boldsymbol{\delta}(z)\,\mathrm{d}z \right) = \left[ -(1-\tau)\,\boldsymbol{X}^\top \int_0^t \boldsymbol{\delta}(z) \right]_+^{\odot\frac{1}{1-\tau}},$$

$$\boldsymbol{w}_-(t) = \exp_\tau \left( -\frac{1}{1-\tau} + \int_0^t \boldsymbol{X}^\top \boldsymbol{\delta}(z)\,\mathrm{d}z \right) = \left[ +(1-\tau)\,\boldsymbol{X}^\top \int_0^t \boldsymbol{\delta}(z) \right]_+^{\odot\frac{1}{1-\tau}}.$$

Setting $\boldsymbol{\lambda} = -(1-\tau) \int_0^\infty \boldsymbol{\delta}(z)$ satisfies the remaining KKT conditions. $\square$

**Corollary 1.** *Under the assumptions of Theorem 4, the reparameterized tempered $EGU_\tau^\pm$ updates (22) also recover the minimum $L_{2-\tau}$-norm solution where $\boldsymbol{w}(t) = q_\tau(\boldsymbol{u}_+(t)) - q_\tau(\boldsymbol{u}_-(t))$.*

This corollary shows that reparameterizing the loss in terms of the parameters $\boldsymbol{u}$ changes the implicit bias of the GD updates. Similar results were observed before in terms of sparse signal recovery [Vaskevicius et al., 2019] and matrix factorization [Gunasekar et al., 2017]. Here, we show that this is a direct result of the dynamics induced by the reparameterization Theorem 2.

## 5 Conclusion and Future Work

In this paper, we motivated the continuous-time mirror descent updates and provided a general framework for reparameterizing these updates. We also introduced the tempered $EGU_\tau^\pm$ updates and its reparameterizations. The tempered $EGU_\tau^\pm$ updates include the two commonly used gradient descent and exponentiated gradient updates, and interpolations between them. For the underdetermined linear regression problem we showed that under certain conditions, the tempered $EGU_\tau^\pm$ updates converge to the minimum $L_{2-\tau}$-norm solution. The current work leads to many interesting future directions:

- The focus is this paper was to develop the reparameterization method in full generality. Our reparameterization equivalence theorem holds only in the continuous-time and the equivalence relation breaks down after discretization. However, in many important cases the discretized reparameterized updates closely track the discretized original updates [Amid and Warmuth, 2020]. This was done by proving the same on-line worst case regret bounds for the discretized reparameterized updates as the originals. A key research direction is to find general conditions for which this is true.

- Perhaps the most important application of the current work is reparameterizing the weights of deep neural networks for achieving sparse solutions or obtaining an implicit form of regularization that mimics a trade-off between the ridge and lasso methods (e.g. elastic net regularization [Zou and Hastie, 2005]). Here the key open question is the following: Are sparse networks (as in Figure 2) necessary, if the goal is to learn sample efficiently when the solution is sparse?[10]

- A general treatment of the convergence results for underdetermined linear regression should allow any start vector. Also, developing a matrix form of the reparameterization theorem is left open.

## Acknowledgement

Thanks to Zakaria Mhammedi for valuable feedback and to NSF grant IIS-1546459 for partial support.

## Broader Impact

The result of the paper suggests that the mirror descent updates can be effectively used in neural networks by running backpropagation on the reparameterized form of the neurons. This may have a potential use case for training these networks more efficiently. This is a theoretical paper and the broader ethical impact discussion is not applicable.

## Footnotes

[3] Whereas the squared Euclidean divergence (which motivates GD) is strongly-convex w.r.t. the $L_2$-norm.

[4] The gradient of a scalar function is a special case of a Jacobian, and should therefore be denoted by a row vector. However, in this paper we use the more common column vector notation for gradients, i.e. $\nabla F(\boldsymbol{w}) \coloneqq \left(\frac{\partial F}{\partial \boldsymbol{w}}\right)^{\top}$.

[5] An equivalent integral form of the CMD update is $\boldsymbol{w}(t) = f^{-1}\Big( f(\boldsymbol{w}_s) - \int_{z=s}^{t} \nabla L(\boldsymbol{w}(z))\, dz \Big)$.

[6] The objective of (3) is essentially a discretization of the objective of (6). See Appendix C.

[7] That is, the value of one variable does not depend on changes in the other.

[8]The second form is more commonly known as $\beta$-divergence [Cichocki and Amari, 2010] with $\beta = 2 - \tau$.

[9]For example, $\tau = 2$ corresponds to the Burg updates (Example 4).

[10]This has recently been proven in Warmuth et al. [2020].

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
