[Supplementary Material]

## A   Dual Form of Bregman Momentum

The dual form of Bregman momentum given in (10) can be obtained by first forming the dual Bregman divergence in terms of the dual variables $\boldsymbol{w}^*(t)$ and $\boldsymbol{w}_0^*$ and then taking the time derivative:

$$
\dot{D}_F(\boldsymbol{w}(t), \boldsymbol{w}_0) = \dot{D}_{F^*}(\boldsymbol{w}_0^*, \boldsymbol{w}^*(t)) = \frac{\partial}{\partial t}\left( F^*(\boldsymbol{w}_0^*) - F^*(\boldsymbol{w}^*(t)) - f^*(\boldsymbol{w}^*(t))^\top \left( \boldsymbol{w}_0^* - \boldsymbol{w}^*(t) \right) \right)
$$

$$
= -\dot{F^*}(\boldsymbol{w}^*(t)) + f^*(\boldsymbol{w}^*(t))^\top \dot{\boldsymbol{w}}^*(t) + (\boldsymbol{w}^*(t) - \boldsymbol{w}_0^*)^\top \boldsymbol{H}_{F^*}(\boldsymbol{w}^*(t)) \ \dot{\boldsymbol{w}}^*(t)
$$

$$
= \left( \boldsymbol{w}^*(t) - \boldsymbol{w}_0^* \right)^\top \boldsymbol{H}_{F^*}(\boldsymbol{w}^*(t)) \ \dot{\boldsymbol{w}}^*(t) \,,
$$

where we use the fact that $\dot{F^*}(\boldsymbol{w}^*(t)) = f^*(\boldsymbol{w}^*(t))^\top \dot{\boldsymbol{w}}^*(t)$.

## B   Constrained Updates and Reparameterization

We first provide a proof for Proposition 1. Then, we prove Theorem 3.

**Proposition 1.** *The CMD update with the additional constraint $\psi\big(\boldsymbol{w}(t)\big) = \boldsymbol{0}$ for some function $\psi : \mathbb{R}^d \to \mathbb{R}^m$ s.t. $\{\boldsymbol{w} \in \mathcal{C} \,|\, \psi\big(\boldsymbol{w}(t)\big) = \boldsymbol{0}\}$ is non-empty, amounts to the projected gradient update*

$$
\dot{f}\big(\boldsymbol{w}(t)\big) = -\boldsymbol{P}_\psi(\boldsymbol{w}(t))\nabla L(\boldsymbol{w}(t)) \ \& \ \dot{f^*}\big(\boldsymbol{w}^*(t)\big) = -\boldsymbol{P}_\psi(\boldsymbol{w}(t))^\top \nabla L \circ f^*\left(\boldsymbol{w}^*(t)\right), \tag{14}
$$

*where $\boldsymbol{P}_\psi \coloneqq \boldsymbol{I}_d - \boldsymbol{J}_\psi^\top \big( \boldsymbol{J}_\psi \boldsymbol{H}_F^{-1} \boldsymbol{J}_\psi^\top \big)^{-1} \boldsymbol{J}_\psi \boldsymbol{H}_F^{-1}$ is the projection matrix onto the tangent space of $F$ at $\boldsymbol{w}(t)$ and $\boldsymbol{J}_\psi(\boldsymbol{w}(t))$. Equivalently, the update can be written as a projected natural gradient descent update*

$$
\dot{\boldsymbol{w}}(t) = -\boldsymbol{P}_\psi^\top (\boldsymbol{w}(t))\boldsymbol{H}_F^{-1}(\boldsymbol{w}(t))\nabla L(\boldsymbol{w}(t)) \ \& \ \dot{\boldsymbol{w}}^*(t) = -\boldsymbol{P}_\psi \boldsymbol{H}_{F^*}^{-1}(\boldsymbol{w}^*(t))\nabla L \circ f^*(\boldsymbol{w}^*(t)). \tag{15}
$$

*Proof of Proposition 1.* We use a Lagrange multiplier $\boldsymbol{\lambda}(t) \in \mathbb{R}^m$ in (6) to enforce the constraint $\psi(\boldsymbol{w}(t)) = \boldsymbol{0}$ for all $t \geq 0$,

$$
\min_{\boldsymbol{w}(t)} \left\{ \dot{D}_F(\boldsymbol{w}(t), \boldsymbol{w}_s) + L(\boldsymbol{w}(t)) + \boldsymbol{\lambda}(t)^\top \psi(\boldsymbol{w}(t)) \right\}. \tag{23}
$$

Setting the derivative w.r.t. $\boldsymbol{w}(t)$ to zero, we have

$$
\dot{f}(\boldsymbol{w}(t)) + \nabla_{\boldsymbol{w}} L(\boldsymbol{w}(t)) + \boldsymbol{J}_\psi(\boldsymbol{w}(t))^\top \boldsymbol{\lambda}(t) = \boldsymbol{0}, \tag{24}
$$

where $\boldsymbol{J}_\psi(\boldsymbol{w}(t))$ is the Jacobian of the function $\psi(\boldsymbol{w}(t))$. In order to solve for $\boldsymbol{\lambda}(t)$, first note that $\dot{\psi}(\boldsymbol{w}(t)) = \boldsymbol{J}_\psi(\boldsymbol{w}(t))\,\dot{\boldsymbol{w}}(t) = \boldsymbol{0}$. Using the equality $\dot{f}(\boldsymbol{w}(t)) = \boldsymbol{H}_F(\boldsymbol{w}(t))\dot{\boldsymbol{w}}(t)$ and multiplying both sides by $\boldsymbol{J}_\psi(\boldsymbol{w}(t))\boldsymbol{H}_F^{-1}(\boldsymbol{w}(t))$ yields (ignoring $t$)

$$
\cancel{\boldsymbol{J}_\psi(\boldsymbol{w})\dot{\boldsymbol{w}}} + \boldsymbol{J}_\psi(\boldsymbol{w})\boldsymbol{H}_F^{-1}(\boldsymbol{w})\nabla L(\boldsymbol{w}) + \boldsymbol{J}_\psi(\boldsymbol{w})\boldsymbol{H}_F^{-1}(\boldsymbol{w})\boldsymbol{J}_\psi^\top(\boldsymbol{w})\boldsymbol{\lambda}(t) = \boldsymbol{0}.
$$

Assuming that the inverse exists, then

$$
\boldsymbol{\lambda} = -\big(\boldsymbol{J}_\psi(\boldsymbol{w})\boldsymbol{H}_F^{-1}(\boldsymbol{w})\boldsymbol{J}_\psi^\top(\boldsymbol{w})\big)^{-1} \boldsymbol{J}_\psi(\boldsymbol{w})\boldsymbol{H}_F^{-1}(\boldsymbol{w})\nabla L(\boldsymbol{w}).
$$

Plugging in for $\boldsymbol{\lambda}(t)$ yields (15). Multiplying both sides by $\boldsymbol{H}_F(\boldsymbol{w})$ and using $\dot{f}(\boldsymbol{w}) = \boldsymbol{H}_F(\boldsymbol{w})\dot{\boldsymbol{w}}$ yields (14). □

**Theorem 3.** *The constrained CMD update (14) coincides with the reparameterized projected gradient update on the composite loss,*

$$
\dot{g}\big(\boldsymbol{u}(t)\big) = -\boldsymbol{P}_{\psi \circ q}(\boldsymbol{u}(t))\nabla_{\boldsymbol{u}} L \circ q(\boldsymbol{u}(t)),
$$

*where $\boldsymbol{P}_{\psi \circ q} \coloneqq \boldsymbol{I}_k - \boldsymbol{J}_{\psi \circ q}^\top \big(\boldsymbol{J}_{\psi \circ q}\boldsymbol{H}_G^{-1}\boldsymbol{J}_{\psi \circ q}^\top\big)^{-1}\boldsymbol{J}_{\psi \circ q}\boldsymbol{H}_G^{-1}$ is the projection matrix onto the tangent space at $\boldsymbol{u}(t)$ and $\boldsymbol{J}_{\psi \circ q}(\boldsymbol{u}) \coloneqq \boldsymbol{J}_q^\top(\boldsymbol{u})\boldsymbol{J}_\psi(\boldsymbol{w}).$*

*Proof of Theorem 3.* Similar to the proof of Proposition 1, we use a Lagrange multiplier $\boldsymbol{\lambda}(t) \in \mathbb{R}^m$ to enforce the constraint $\psi \circ q(\boldsymbol{u}(t)) = \mathbf{0}$ for all $t \geq 0$,

$$\min_{\boldsymbol{u}(t)} \left\{ \dot{D}_G(\boldsymbol{u}(t), \boldsymbol{u}_s) + L \circ q(\boldsymbol{u}(t)) + \boldsymbol{\lambda}(t)^\top \psi \circ q(\boldsymbol{u}(t)) \right\}.$$

Setting the derivative w.r.t. $\boldsymbol{u}(t)$ to zero, we have

$$\dot{g}(\boldsymbol{w}(t)) + \nabla_{\boldsymbol{u}} L \circ q(\boldsymbol{w}(t)) + \boldsymbol{J}_{\psi \circ q}^\top(\boldsymbol{u}(t)) \boldsymbol{\lambda}(t) = \mathbf{0},$$

where $\boldsymbol{J}_{\psi \circ q}(\boldsymbol{u}(t)) := \boldsymbol{J}_q^\top(\boldsymbol{u}) \nabla \psi(\boldsymbol{w}(t))$. In order to solve for $\boldsymbol{\lambda}(t)$, we use the fact that $\dot{\psi} \circ q(\boldsymbol{u}(t)) = \boldsymbol{J}_{\psi \circ q}(\boldsymbol{u}(t)) \dot{\boldsymbol{u}}(t) = \mathbf{0}$. Using the equality $\dot{g}(\boldsymbol{u}(t)) = \boldsymbol{H}_G(\boldsymbol{u}(t)) \dot{\boldsymbol{u}}(t)$ and multiplying both sides by $\boldsymbol{J}_{\psi \circ q}(\boldsymbol{u}(t)) \boldsymbol{H}_G^{-1}(\boldsymbol{u}(t))$ yields (ignoring $t$)

$$\boldsymbol{J}_{\psi \circ q}(\boldsymbol{u}) \dot{\boldsymbol{u}} + \boldsymbol{J}_{\psi \circ q}(\boldsymbol{w}) \boldsymbol{H}_G^{-1}(\boldsymbol{u}) \nabla L \circ q(\boldsymbol{u}) + \boldsymbol{J}_{\psi \circ q}(\boldsymbol{w}) \boldsymbol{H}_G^{-1}(\boldsymbol{w}) \boldsymbol{J}_{\psi \circ q}^\top(\boldsymbol{u}) \boldsymbol{\lambda}(t) = \mathbf{0}.$$

The rest of the proof follows similarly by solving for $\boldsymbol{\lambda}(t)$ and rearranging the terms. Finally, applying the results of Theorem 2 concludes the proof. $\square$

## C Discretized Updates

In this section, we discuss different strategies for discretizing the CMD updates and provide examples for each case.

The most straight-forward discretization of the unconstrained CMD update (1) is the forward Euler (i.e. explicit) discretization, given in (5). Note that this corresponds to a minimizer of the discretized form of (6) with a step size of $h$, except that the initial weight vector is $\boldsymbol{w}_s$ instead of $\boldsymbol{w}_0$. That is,

$$\underset{\boldsymbol{w}}{\operatorname{argmin}} \left\{ 1/h \left( D_F(\boldsymbol{w}, \boldsymbol{w}_s) - \underbrace{D_F(\boldsymbol{w}_s, \boldsymbol{w}_s)}_{=0} \right) + L(\boldsymbol{w}) \right\}.$$

An alternative way of discretizing is to apply the approximation on the equivalent natural gradient form (11), which yields

$$\boldsymbol{w}_{s+1} - \boldsymbol{w}_s = -h \, \boldsymbol{H}_F^{-1}(\boldsymbol{w}_s) \, \nabla L(\boldsymbol{w}_s).$$

Despite being equivalent in continuous-time, the two approximations may correspond to different updates after discretization. As an example, for the EGU update motivated by $f(\boldsymbol{w}) = \log \boldsymbol{w}$ link, the latter approximation yields

$$\boldsymbol{w}_{s+1} = \boldsymbol{w}_s \odot \left( \mathbf{1} - h \, \nabla L(\boldsymbol{w}_s) \right),$$

which amounts to approximating the exponential factor $\exp(-\eta \nabla L(\boldsymbol{w}_s)$ in the EGU update by its Taylor expansion $(\mathbf{1} - h \, \nabla L(\boldsymbol{w}_s))$.

The situation becomes more involved for discretizing the constrained updates. As the first approach, it is possible to directly discretize the projected CMD update (14)

$$f(\widetilde{\boldsymbol{w}}_{s+1}) - f(\boldsymbol{w}_s) = -h \, \boldsymbol{P}_\psi(\boldsymbol{w}_s) \nabla L(\boldsymbol{w}_s).$$

However, note that the new parameter $\widetilde{\boldsymbol{w}}_{w+1}$ may fall outside the constraint set $\mathcal{C}_\psi := \{\boldsymbol{w} \in \mathcal{C} \,|\, \psi(\boldsymbol{w})) = \mathbf{0}\}$. As a result, a Bregman projection [Shalev-Shwartz et al., 2012] into $\mathcal{C}_\psi$ may need to be applied after the update, that is

$$\boldsymbol{w}_{s+1} = \underset{\boldsymbol{w} \in \mathcal{C}_\psi}{\operatorname{argmin}} D_F(\boldsymbol{w}, \widetilde{\boldsymbol{w}}_{s+1}). \tag{25}$$

As an example, for the normalized EG updates with the additional constraint that $\boldsymbol{w}^\top \mathbf{1} = 1$, we have $\boldsymbol{P}_\psi(\boldsymbol{w}) = \boldsymbol{I}_d - \mathbf{1} \boldsymbol{w}^\top$ and the approximation yields

$$\log(\widetilde{\boldsymbol{w}}_{s+1}) - \log(\boldsymbol{w}_s) = -h \left( \nabla L(\boldsymbol{w}_s) - \mathbf{1} \, \mathbb{E}_{\boldsymbol{w}_s}[\nabla L(\boldsymbol{w}_s)] \right),$$

where $\mathbb{E}_{\boldsymbol{w}_s}[\nabla L(\boldsymbol{w}_s)] = \boldsymbol{w}_s^\top \nabla L(\boldsymbol{w}_s)$. Clearly, $\widetilde{\boldsymbol{w}}_{s+1}$ may not necessarily satisfy $\widetilde{\boldsymbol{w}}_{s+1}^\top \mathbf{1} = 1$. Therefore, we apply

$$\boldsymbol{w}_{s+1} = \frac{\widetilde{\boldsymbol{w}}_{s+1}}{\|\widetilde{\boldsymbol{w}}_{s+1}\|_1},$$

which corresponds to the Bregman projection onto the unit simplex using the relative entropy divergence [Kivinen and Warmuth, 1997].

An alternative approach for discretizing the constrained update would be to first discretize the functional objective with the Lagrange multiplier (23) and then (approximately) solve for the update. That is,

$$\boldsymbol{w}_{s+1} = \operatorname*{argmin}_{\boldsymbol{w}} \left\{ 1/h \left( D_F(\boldsymbol{w}, \boldsymbol{w}_s) - \underbrace{D_F(\boldsymbol{w}_s, \boldsymbol{w}_s)}_{=0} \right) + L(\boldsymbol{w}) + \boldsymbol{\lambda}^\top \psi(\boldsymbol{w}) \right\}.$$

Note that in this case, the update satisfies the constraint $\psi(\boldsymbol{w}_{s+1}) = \mathbf{0}$ because of directly using the Lagrange multiplier. For the normalized EG update, this corresponds to the original normalized EG update in [Littlestone and Warmuth, 1994],

$$\boldsymbol{w}_{s+1} = \frac{\boldsymbol{w}_s \odot \exp\left( - h \, \nabla L(\boldsymbol{w}_s) \right)}{\| \boldsymbol{w}_s \odot \exp\left( - h \, \nabla L(\boldsymbol{w}_s) \right) \|_1} \,.$$

Finally, it is also possible to discretized the projected natural gradient update (15). Again, a Bregman projection into $\mathcal{C}_\psi$ may need to be required after the update, that is,

$$\widetilde{\boldsymbol{w}}_{s+1} - \boldsymbol{w}_s = -h \, \boldsymbol{P}_\psi(\boldsymbol{w}_s)^\top \boldsymbol{H}_F^{-1}(\boldsymbol{w}_s) \nabla L(\boldsymbol{w}(t)) \,,$$

followed by (25). For the normalized EG update, the first step corresponds to

$$\boldsymbol{w}_{s+1} = \boldsymbol{w}_s \odot \left( \mathbf{1} - h \big( \nabla L(\boldsymbol{w}_s) - \mathbf{1} \, \mathbb{E}_{\boldsymbol{w}_s}[\nabla L(\boldsymbol{w}_s)] \big) \right),$$

which recovers to the *approximated EG* update of Kivinen and Warmuth [1997]. Note that $\boldsymbol{w}_{s+1}^\top \mathbf{1} = 1$ and therefore, no projection step is required in this case.