[Reviews · NeurIPS 2020]

Review 1

Summary and Contributions: This paper presents a method for using a different gradient scheme for optimization. This scheme, mirror descent, is based on Bergman divergence, which can offer a potential advantage to the convergence of certain systems. Neural networks typically use gradient descent based on quadratic loss and the proposed method could offer an advantage to neural network training. Namely, the authors point out that, with small neural networks, training convergence is provably faster with this method. The authors’ proposal is a general framework for casting mirror descent as gradient descent via reparameterization. When later applied to neural network training, this method could lead to large advantages in learning from data faster and cheaper with current compute systems.

Strengths: The theoretical grounding is well founded from the optimization literature and is well motivated by the difficulty and expense in training large neural networks. This could be on the path to fundamental breakthroughs in our field that allow us to approach the capacity and efficiency of biological brains. To my knowledge, I haven’t seen a generalized framework such as this, so it is novel work.

Weaknesses: The authors do a good job of listing some of the future directions to which this work could lead. I believe the authors could relate this work more directly to neural networks, as this is a timely subject. In this vein, seeing where the assumptions of casting the problem as Mirror descent breakdown in relation to neural networks would be of high value.

Correctness: To the best of my knowledge, the derivations are correct. However, I am not well equipped to critique each step of them.

Clarity: Generally, a well written paper. The authors assume the readers are well versed in the background of this work. As I understand it is difficult to provide adequate background and include all of the derivations required to motivate the result, I would have appreciated more context around each step. For instance, what is the intuition around mirror descent and Bergman divergence? Why is using MD vs GD potentially better? I believe there needs to be more background to motivate the need for such work and then apply it to a problem that the field works on directly.

Relation to Prior Work: This is part of a set of work around MD, EGU, and Bergman divergence. To my reading, the appropriate background is cited, but I am not familiar enough with the background to identify missing work. I was able to read some of the references and get a sense of what lead to this work.

Reproducibility: Yes

Additional Feedback: This is a theoretical result. There aren’t any experiments to reproduce, but researchers in this field should be able to follow the logic and identify any errors. I acknowledge any rebuttals and stand by my review. The current review has been updated based on rebuttals.


Review 2

Summary and Contributions: The paper provides a framework to convert a mirror descent algorithm into a gradient descent (GD) algorithm through a reparametrization of the parameter to be learned. The mirror descent considered in the paper is the continuous-time version, i.e., continuous mirror descent (CMD). The paper then presents several examples of CMD and the corresponding GD to demonstrate the idea.

Strengths: The claim of result is clear. Main results are well-demonstrated with several examples. Converting CMD to GD is an interesting topic and is closely related to the NeurIPS community.

Weaknesses: The writing: The writing of the paper gave me some difficulty in understanding it. There is lack of definition. Some symbols and expressions are used without clearly defined. Problems is refered to without definition. Transition between parts/sections requires improvement. Please see 'Additional feedback' for more detail.

Correctness: The claim and method look correct to me.

Clarity: There is still room for improvement.

Relation to Prior Work: Yes, it is discussed.

Reproducibility: Yes

Additional Feedback: Suggestions: Lack definition (anything that is not 'common knowledge' should be defined and explained before using. Should not let readers guess.) 1. In eq(1), w and L is used without defined. Could first introduce the problem and mention L is loss or the target function, and w is the model parameter. 2. In theorem 2 line 116 the phrase 'coincides with' is unclear. 'coincides with' is not a commonly used, mathematically rigorous and clear expression. So if authors want to use it, please define clearly what it means before using. Although I can guess the meaning and finally understand what thm2 is saying after reading the paper, the theorems should be self-contained and claims be mathematically clear. 3. In section 4.1, under-determined linear regression problem is not properly defined before being refered to. What is L, w and corresponding F,f in this case? 4. line 83: what is the meaning of 'independent variable' in this setting? Connection between sections: 1. The paper discuss the dual form of the problem in between line 87 and 103. When I first read this part, I am not sure why the authors put it here. There is no motivation of this part, and no connection between this part and the previous part, making the transition unsmooth. I recommend add some motivation before line 87 so the trasition is smooth. It is even ok to make this part into a subsection itself, with motivation and explanation, since it has importance: the equivalence is a special case of the main result; eq(11) is used in the proof of theorem 2. Questions: 1. Why should we care about reparametrize CMD as GD? What's the benefit (eg. fast training, better interpretation)?


Review 3

Summary and Contributions: This paper studies reparameterization method for the continuous-time mirror descent algorithm. The authors firstly prove that, similar to MD, CMD can be presented as minimizing a tradeoff between the loss and a Bregman momentum. They then provide a general reparameterization theorem, which allows reparametrizing one CMD update by another. Based on this technique, they further show that various CMD updates including EGU, Burg updates and EG can be reparametrized as GD update. Finally, the authors study tempered updates, which allow interpolating between different updates.

Strengths: 1. Gradient descent is widely used in real-world applications, while CMD can be more efficient for many different classes of problems. This paper studies how to reparametrize CMD as GD updates, which is an interesting and well-motived optimization problem. 2. The authors successfully develop a general framework that reals the connections between different CMD updates, and show that how to reparametrize a variety of CMD methods as GD updates.

Weaknesses: In section 2, the authors show that CMD can be presented as minimizing a tradeoff between the loss and a Bregman momentum. Although it is novel and very intresting, I am unsure about the significance of this contribution. It would be better if the authors can add more discussion on this point.

Correctness: I have read the main paper and I didn’t find any significant errors.

Clarity: The paper is generally well-written.

Relation to Prior Work: The relation to prior work is clearly discussed .

Reproducibility: Yes

Additional Feedback:

[Author Response · NeurIPS 2020]

Thanks for the helpful feedback. We got a clear sense of where more clarification would be helpful.

**General Discussion** Our work is part of the following larger and important discussion within the NeurIPS community:
To what solution do neural nets (trained w. GD) converge to in the overparameterized setting? This discussion was
reignited by two recent NeurIPS papers (Gunasekar et al. 2017) and (Vaskevicius et al. 2019). They showed that in a
certain context, continuous GD updates converge to sparse solutions. In a recent COLT paper (Amid, Warmuth 2020)
this was related to a particular MD algorithm: if the edges of a linear neuron are doubled (as in Fig. 2), then continuous
GD on this network simulates the unnormalized exponentiated gradient algorithm (EGU). EGU is the classical MD
algorithm (with the log link) and there is a long history of results showing that it does well when the solution is sparse.
Previously it was thought that GD cannot take advantage of the sparsity of the solution.

In this paper we show that the reach of GD is even much farther: Any CMD update can be simulated by another CMD
provided a certain reparameterization function exists. We give many reparameterization examples in the paper and in
particular, we develop a family of MD updates (using the $\log_\tau$ link) that lie between GD and EGU using vanilla GD
(by means of a reparameterization).

What is the surprising insight? The structure of the network is key! Much research has been done in exploring multi
layer nets with fully connected (FC) layers. In particular, wide FC layers have been investigated. For example the
NeurIPS paper, Arora et al. "On Exact Computation with an Infinitely Wide Neural Net." NeurIPS 19, shows that an
infinitely wide neural net behaves like a kernel method.

Our research leads us in a different direction in that a linear neuron in which the edges are doubled (Fig. 2) can exploit
sparsity. In particular, combined with some well known bounds for any kernel method, this shows that GD on the thin
network of Fig. 2. can learn certain sparse problems exponentially faster than any kernel method. We believe that this
discussion is central to neural net research and important to the NeurIPS community.

We will add more context along the above to the final version. We will also smoothen the transitions and give more
extensive definitions as suggested by the reviewers. The discussion of duality is important for the optimization context.
We will relegate much of that to the appendix in exchange for giving more motivation for the main results of the paper.

**Reviewer #2** • *"given that this work is being submitted to NeurIPS, I don't see enough of a relationship to artificial*
*neural networks. . . Describing the specifics of using this work to help the neural processing world feels as though it*
*would be more appropriate."*

Please note that many similar theoretical work such as (Gunasekar et al. 2017) and (Vaskevicius et al. 2019)
have appeared before in NeurIPS. That is, NeurIPS has always been one of the top venues for publishing results on
learning theory and optimization. The current submission is a theoretical work, but also has practical implications for
understanding the training dynamics of deep neural networks. Therefore, we would like to encourage Reviewer #2 to
cast their judgment solely based on the quality of the paper and significance of the theoretical results and defer the
concerns about the relevance of the work to the NeurIPS community to the ACs.

**Reviewer #4** • *Lack of definitions:* We will add the missing definitions and improve the flow.

• *Meaning of 'independent variables:* We refer to independence by means of elementary calculus. For independent
variables, a variable in an equation may have its value freely chosen without considering values of the other variables.

• *Motivation of the dual form:* The dual form of MD is extremely important for efficient implementation of updates
and analyzing the worst case regret bounds. We will make the motivation clear. We will also improve the write up to
ease the transition between the sections.

• *"Why should we care about reparameterized CMD as GD?"*

For a long time, many MD updates such as EGU had been considered fundamentally different than GD (multiplicative
vs. additive). Therefore, the EGU update was considered to be unrealizable by additive updates such as backprop. We
show that EGU (and many other CMD) updates are realizable "exactly" via backprop on a reparameterized network.
Following (Amid, Warmuth 2020), we believe that the discretized updates will closely mimic the original updates.
This has important implications for implementing these updates via GD in the existing platforms (such as TF).

**Reviewer #5** • *"the authors show that CMD can be presented as minimizing a trade-off between the loss and a*
*Bregman momentum. ... I am unsure about the significance of this contribution."*

Prior to this work, CMD was motivated as the limit point of MD when the step-size goes to zero. Alternatively, we
motivate CMD as the minimization of a functional objective. This novel view of CMD is useful for deriving worst case
regret bounds in the continuous-time, which we defer to future work. This also allows a novel approach for deriving
MD updates by directly discretizing the functional objective (please see Appendix C for more details).

[Meta-Review · NeurIPS 2020]

The theoretical result of the paper is significant in my opinion. I also agree with the authors that the topic of this paper is at the core of machine learning and thus the paper should be evaluated based on its contributions. The reviewers also adjusted their reviews based on this point. However, I should also mention that the reviewers raised the concern that some of the definitions are omitted and few parts of the paper is not rigorous enough. Therefore, I suggest that the authors take care of such ambiguities in the final version.